# Comparison of intra-session reliability of force-velocity-power variables between a horizontal dynamic leg press device and vertical jump tests

**Takuya Nishioka**[1], **Shota Yamaguchi**[1], **Daiki Hajima**[2], **Yasuhiro Kunita**[2], **Takayuki Inami**[1,2*]

**1** Institute of Physical Education, Keio University, Yokohama, Japan, **2** Graduate School of Health Management (Sports and Health Sciences), Keio University, Yokohama, Japan

* inamit@keio.jp

## Abstract

This study aimed to compare the intra-session reliability of force-velocity-power variables obtained from a horizontal dynamic leg press device (HDLPD) and vertical jump tests. Nineteen male field hockey players performed maximal leg extension movements in HDLPD under a wide range of available load conditions (100, 120, 140, 160, 180, and 200% body weight [BW]), followed by squat jumps (SJ) and countermovement jumps (CMJ) under unloaded (0 kg), moderate- (22.0 ± 4.6 kg), and high-loaded (43.9 ± 9.2 kg) conditions. The peak and mean values of force, velocity, and power derived from the HDLPD, SJ, and CMJ were calculated. The HDLPD showed acceptable intrasession reliability for all experimental outcomes (intraclass correlation coefficient [ICC] = 0.845 to 0.974, coefficient of variation [CV] = 0.43% to 4.52%). The outcome variables during SJ and CMJ had acceptable intra-session reliability, except for the relative reliability of some variables (ICC = 0.588–0.973, CV = 1.19%–8.38%). The results of this study showed that the HDLPD has high intrasession reliability in measuring force-velocity-power variables for maximal leg extension performance. In addition, for some force-velocity-power variables, HDLPD can provide more reliable measurements than vertical jumps. Therefore, the HDLPD can be useful for practitioners who want to reliably measure leg extension strength, speed, and power outcomes.

## Introduction

Leg extension is a fundamental movement performed in numerous athletic activities, including jumping, sprinting, and change of direction [1]. Reliable measurement of kinetic and kinematic variables – such as force, velocity, and power – during leg extension allows the accurate detection of training-induced performance changes [2].

**Data availability statement:** All relevant data are within the manuscript and its Supporting Information files. Some more detailed data is available from inamit@keio.jp.

**Funding:** N.I.S. New Investment Solutions (Liechtenstein) AG funded and Dynamic Devices Co., Ltd. collaborated with this study; however, the company did not influence the design of the study, the collection, analysis, and scientific interpretation of the collected data, or the writing of the manuscript, nor the publication of this article.

**Competing interests:** The authors have declared that no competing interests exist.

Consequently, the reliability of leg extension performance measurements has been the focus of many previous studies [3–5].

Vertical jumping is one of the most commonly used leg extension performance measurements [6]. Most previous studies [5,7] have reported acceptable reliability of the kinetic and kinematic variables obtained from vertical jumps. Furthermore, vertical jump measurements are widely used in the field because they are inexpensive and easy to perform [7]. However, in the landing motion immediately after a vertical jump, the subject is exposed to a large eccentric loading and peak vertical ground reaction force (approximately 4–5 times their body weight) [8,9]. Therefore, in cases where minimization of injury risk or muscle damage is required, or if the subject is in rehabilitation, the vertical jump may not be the best option for measuring leg extension capacity.

In recent years, the horizontal dynamic leg press device (HDLPD) has been used as an alternative method to measure leg extension performance [3,4,10,11]. The HDLPD can measure leg extension force, velocity, and power variables with a concentric-only action without landing movements or large eccentric loading [3]. Therefore, the HDLPD can address the above-mentioned problems associated with vertical jump measurements. In addition, some previous studies [3,4] report that the HDLPD has acceptable intersession reliability for measuring leg extension capacity. However, to the best of our knowledge, no studies have reported on the intrasession reliability of the HDLPD. It is unknown whether its reliability is higher than that of the common vertical jump measurement. Intrasession reliability is important in research and clinical practice, particularly for accurately detecting differences between individuals.

Therefore, this study aimed to investigate the intrasession reliability of force-velocity-power variables obtained from the HDLPD and vertical jumps. We hypothesized that 1) HDLPD would have high intra-session reliability in measuring force-velocity-power variables, and 2) the intra-session reliability of force-velocity-power variables obtained from HDLPD would be higher than those obtained from vertical jumps.

## Materials and methods

### Experimental design

A repeated-measures design was used to assess the relative and absolute reliability of a test of peak and mean force-velocity-power variables obtained from an HDLPD and two common types of vertical jumps (squat jump [SJ] and countermovement jump [CMJ]) using a wide range of available loads. A familiarization session for the experimental trials was conducted on the first day, and the main performance measurement was performed on the second day 48–144 h later.

### Participants

This study was conducted in Japan from December 16, 2022, to March 31, 2024. Sample size calculations were performed using G*Power software (version 3.1.9.6, Düsseldorf, Germany) to determine the number of participants required to assess reliability [12]. Based on a detected reliability of 0.9, a minimal acceptable reliability

of 0.5, a significance level of 0.05, and a power of 0.95, the required sample size was calculated to be 17 participants. Accordingly, 19 male field hockey players (age: 19.6 ± 1.1 years, height: 173.0 ± 5.9 cm, body mass [BM]: 61.8 ± 6.5 kg, mean ± SD) were recruited to participate in this study; they were free of any musculoskeletal pain or injury that could compromise testing. None of the participants was under 18 years of age. The participants were instructed to avoid any strenuous exercise 24 h prior to the experiments. After explaining the purpose of the study, its procedures, risks, and benefits, written informed consent was obtained prior to participation. This study was conducted in accordance with the principles of the Declaration of Helsinki and approved by the ethics committee of our university (approval number: 22−012; approved on December 16, 2022).

### Familiarization and performance measurement sessions

On the first day, the participants were familiarized with the experimental trials using a dynamic leg press device, as well as unloaded and loaded vertical jumps (SJ and CMJ), while gradually increasing the external loads. In this familiarization session, the external load that made the participant's SJ height 10–15 cm was recorded and used as the load in the high-loaded condition of the SJ and CMJ for each participant [13]. On the second day, the participants completed a standard 10-min warm-up consisting of jogging, lower body dynamic stretches, SJ, and CMJ. Subsequently, they performed vertical jumps on dual force plates (PASCO, PS-3229, Roseville, USA) [14]. The vertical jump test consisted of the following sequences: unloaded SJ and CMJ, moderate-loaded SJ and CMJ, and high-loaded SJ and CMJ. Both SJ and CMJ were performed twice each under each loading condition, with SJ followed by CMJ. Adequate rest intervals (2–3 min) were provided between jumps.

### Testing procedures

The leg press test was conducted using a HDLPD equipped with pneumatic artificial muscles (ddrobotec® System ELITE mk4, Dynamic Devices, Zurich, Switzerland). Participants began with two familiarization trials at the lightest load, corresponding to 100% of the participant's body weight (BW). Thereafter, the load was gradually increased in fixed steps of 20% BW from 100% BW up to a maximum of 200% BW. In our pilot testing, all participants successfully performed the leg press at 200% BW. Furthermore, this load range corresponds to the forces typically observed during vertical jumps [3]; therefore, 100–200% BW loads were selected for the HDLPD in the present study. Participants performed two trials for each load. Rest periods were progressively lengthened with increasing load: for the first three load levels, rest intervals ranged from 10 to 20 seconds, and for the final three loads, from 20 to 40 seconds. The seat position was individually adjusted to achieve a knee flexion angle of approximately 90°, with both feet placed symmetrically on the footplates. The participants were instructed to extend both legs as forcefully and rapidly as possible during each trial. Owing to the pneumatic semi-isotonic resistance, maximal effort did not result in ballistic action, and the entire push-off was performed with maximal intentional velocity. The leg press was performed as a concentric-only action without a preceding countermovement, as the pedals rested in their consistent starting position prior to each repetition. The eccentric phase was passively guided by the HDLP and was not recorded.

For the SJ, the subjects were instructed to lower themselves into a half-squat position at a knee flexion angle of 90° and hold this position for 2 seconds. Subsequently, they jumped as high as possible without performing a countermovement. The waveform data obtained from the force plates during SJ were checked to confirm that no previous countermovement was used. For the CMJ, the participants were instructed to perform a downward movement as deep as the SJ, perform a countermovement as fast as possible, and jump as high as possible. To control the push-off distance during the SJ and CMJ as much as possible, a timing gate (VoltOnoSprint, S-CADE. Corp., Tokyo, Japan) was placed such that a beep sound was activated when the subject was in a half-squat position at a knee angle of 90°. If the beeping sound from the timing gate was not activated while jumping, the trial was repeated. The subjects performed free-weight SJ and CMJ with a 0.4-kg wooden bar under the unloaded condition, a barbell with the load determined on the first day under the

high-loaded condition (43.9 ± 9.2 kg [71.6 ± 15.5% BM]), and a barbell with half that load under the moderate-loaded condition (22.0 ± 4.6 kg [35.8 ± 7.8% BM]).

## Data analyses

For the leg press assessments, the ddrobotec® System ELITE recorded the force and footplate displacement data (i.e., arc length) at a sampling frequency of 200 Hz, using the integrated pressure sensors and position transducers of the HDLPD. Based on system-specific pre-calibration data, the displacement of the footplates was used to estimate knee joint angles for each participant. The instant when the estimated knee joint angles began to move in the direction of extension for both legs was defined as the initial range of the motion, and the instant when the estimated knee joint angle of the leg decreased to <10° was defined as the leg's end range of motion. Force values were computed as the sum of the values measured for both pedals. The velocity values were calculated by dividing the arc length of the footplate covered by the range of motion by its completion time. The average value for both legs was used for the analyses. Power was calculated as the product of force and velocity.

For SJ and CMJ, the vertical ground reaction force data for all jumps were sampled at a frequency of 1,000 Hz. Signals from the force plate were filtered using a fourth-order Butterworth low-pass filter with a 50-Hz cutoff [15]. Before each jump, the subjects were weighed for 3 seconds to measure the total system weight. The movement initiation of the jump was defined as the time point 30 ms before the vertical ground reaction force exceeded the threshold (the system weight ± 5 SD) [16,17]. For each jump, the center of mass (COM) velocity was calculated using the trapezoid rule, whereas the net vertical ground reaction force was calculated as the force exceeding the system weight divided by the system mass to determine acceleration [16]. Acceleration was numerically integrated to provide instantaneous COM velocity, which was then numerically integrated to provide instantaneous COM displacement [16]. The takeoff and landing thresholds were identified from 5 SD during the flight phase across a 0.3 s (0.1 s if the flight time was <0.3 s) period based on a previous study [18]. The takeoff instances were identified as the first force values greater than the force threshold [18].

## Statistical analyses

Statistical analyses were performed using the Statistical Package for the Social Sciences (SPSS software version 29, IBM Corp., Armonk, NY, USA) and Microsoft Excel spreadsheet [19]. The intraclass correlation coefficient (ICC; two-way mixed effects, absolute agreement, and single observer/measurement) [20], coefficient of variation (CV), standard error of measurement (SEM) [21], and typical error (TE) [19] were calculated to analyze the measurement reliability. Correlation strength was interpreted as poor, moderate, good, and excellent for ICC values of 0.00–0.50, 0.50–0.75, 0.75–0.90, 0.90–1.00, respectively [20]. ICCs > 0.75 and CVs < 10% were considered acceptable [20,22]. For ICC and CV, 95% confidence intervals (CIs) were calculated. To determine the sensitivity of each variable, TE was compared to small and moderate worthwhile change (SWC and MWC, respectively). The SWC and MWC were calculated by multiplying the between-subject SD by 0.2 and 0.6, respectively [23]. If the TE was less than the SWC, the test variable was rated as 'high'; if the TE was higher than the SWC and less than the MWC, it was rated as 'moderate,' and if the TE was higher than the MWC, it was rated as 'low' [23].

## Results

The descriptive statistics (mean ± SD) and intra-session reliability of force-velocity-power variables obtained from the HDLPD and vertical jumps are shown in Tables 1 and 2.

## Discussion

This study was conducted to verify the hypotheses that (1) HDLPD would have high intra-session reliability in measuring force-velocity-power variables, and (2) the intra-session reliability of force-velocity-power variables obtained from HDLPD would be higher than those obtained from vertical jumps. The main findings of this study indicate that the HDLPD

**Table 1. Intra-session reliability of force-velocity-power variables obtained from a horizontal dynamic leg press device.**

| Load (% BW) | Type of value | Variables | Trial 1 | Trial 2 | ICC (95% CI) | CV (95% CI) | SEM | TE | SWC | MWC | Sensitivity |
|---|---|---|---|---|---|---|---|---|---|---|---|
| 100 | Peak | Force (N) | 1269±155 | 1259±163 | 0.942 (0.858 to 0.977) | 2.58 (1.95 to 3.82) | 38.35 | 38.60 | 31.85 | 95.56 | moderate |
| | | Velocity (m/s) | 1.64±0.16 | 1.62±0.17 | 0.923 (0.815 to 0.969) | 2.24 (1.69 to 3.31) | 0.044 | 0.044 | 0.032 | 0.10 | moderate |
| | | Power (W) | 955±193 | 938±201 | 0.946 (0.868 to 0.979) | 4.39 (3.31 to 6.49) | 45.74 | 45.45 | 39.37 | 118.11 | moderate |
| | Mean | Force (N) | 955±104 | 952±110 | 0.954 (0.924 to 0.972) | 1.61 (1.21 to 2.38) | 22.92 | 19.37 | 21.38 | 64.13 | high |
| | | Velocity (m/s) | 1.14±0.12 | 1.13±0.13 | 0.927 (0.881 to 0.956) | 2.92 (2.21 to 4.32) | 0.034 | 0.043 | 0.025 | 0.08 | moderate |
| | | Power (W) | 1101±223 | 1091±241 | 0.936 (0.842 to 0.975) | 4.52 (3.42 to 6.69) | 58.66 | 59.99 | 46.38 | 139.13 | moderate |
| 120 | Peak | Force (N) | 1334±151 | 1327±170 | 0.952 (0.880 to 0.981) | 2.11 (1.59 to 3.12) | 35.07 | 35.86 | 32.02 | 96.05 | moderate |
| | | Velocity (m/s) | 1.57±0.13 | 1.57±0.15 | 0.933 (0.835 to 0.974) | 1.83 (1.38 to 2.72) | 0.035 | 0.037 | 0.028 | 0.08 | moderate |
| | | Power (W) | 966±178 | 963±202 | 0.951 (0.878 to 0.981) | 3.50 (2.64 to 5.18) | 41.96 | 42.94 | 37.91 | 113.74 | moderate |
| | Mean | Force (N) | 1051±113 | 1048±120 | 0.974 (0.956 to 0.998) | 0.95 (0.72 to 1.41) | 18.73 | 12.88 | 23.24 | 69.72 | high |
| | | Velocity (m/s) | 1.10±0.11 | 1.10±0.12 | 0.946 (0.911 to 0.968) | 2.36 (1.78 to 3.49) | 0.025 | 0.034 | 0.022 | 0.07 | moderate |
| | | Power (W) | 1167±226 | 1160±243 | 0.960 (0.901 to 0.985) | 3.19 (2.41 to 4.72) | 46.88 | 47.57 | 46.89 | 140.66 | moderate |
| 140 | Peak | Force (N) | 1387±144 | 1397±148 | 0.914 (0.793 to 0.966) | 2.41 (1.82 to 3.56) | 42.73 | 43.25 | 29.15 | 87.44 | moderate |
| | | Velocity (m/s) | 1.49±0.12 | 1.51±0.12 | 0.856 (0.669 to 0.942) | 2.43 (1.84 to 3.60) | 0.044 | 0.044 | 0.023 | 0.07 | moderate |
| | | Power (W) | 968±166 | 980±169 | 0.914 (0.793 to 0.966) | 4.15 (3.14 to 6.14) | 49.20 | 49.88 | 33.56 | 100.67 | moderate |
| | Mean | Force (N) | 1140±116 | 1138±116 | 0.967 (0.945 to 0.980) | 1.01 (0.76 to 1.50) | 21.11 | 13.80 | 23.24 | 69.73 | high |
| | | Velocity (m/s) | 1.04±0.09 | 1.03±0.09 | 0.943 (0.906 to 0.966) | 2.79 (2.11 to 4.13) | 0.020 | 0.033 | 0.017 | 0.05 | moderate |
| | | Power (W) | 1187±206 | 1180±203 | 0.939 (0.849 to 0.976) | 3.80 (2.87 to 5.63) | 50.41 | 51.54 | 40.83 | 122.48 | moderate |
| 160 | Peak | Force (N) | 1438±176 | 1445±173 | 0.964 (0.910 to 0.986) | 1.94 (1.47 to 2.88) | 33.05 | 33.65 | 34.84 | 104.53 | high |
| | | Velocity (m/s) | 1.42±0.12 | 1.42±0.12 | 0.913 (0.789 to 0.965) | 1.99 (1.50 to 2.94) | 0.035 | 0.035 | 0.024 | 0.07 | moderate |
| | | Power (W) | 964±192 | 972±191 | 0.958 (0.896 to 0.984) | 3.41 (2.57 to 5.04) | 39.23 | 39.79 | 38.29 | 114.86 | moderate |
| | Mean | Force (N) | 1225±132 | 1228±132 | 0.972 (0.953 to 0.983) | 0.69 (0.52 to 1.02) | 22.09 | 10.20 | 26.41 | 79.23 | high |
| | | Velocity (m/s) | 0.97±0.09 | 0.97±0.09 | 0.955 (0.925 to 0.973) | 2.05 (1.55 to 3.03) | 0.019 | 0.025 | 0.018 | 0.05 | moderate |
| | | Power (W) | 1193±229 | 1195±223 | 0.969 (0.922 to 0.988) | 2.67 (2.01 to 3.95) | 39.72 | 40.50 | 45.13 | 135.38 | high |
| 180 | Peak | Force (N) | 1512±165 | 1518±161 | 0.969 (0.922 to 0.988) | 1.49 (1.12 to 2.20) | 28.72 | 29.16 | 32.63 | 97.88 | high |
| | | Velocity (m/s) | 1.37±0.10 | 1.37±0.09 | 0.890 (0.737 to 0.956) | 1.89 (1.42 to 2.79) | 0.031 | 0.033 | 0.019 | 0.06 | moderate |
| | | Power (W) | 987±176 | 991±166 | 0.953 (0.883 to 0.982) | 2.97 (2.24 to 4.39) | 37.12 | 37.96 | 34.25 | 102.74 | moderate |
| | Mean | Force (N) | 1316±139 | 1318±136 | 0.967 (0.939 to 0.982) | 0.43 (0.32 to 0.63) | 24.96 | 7.89 | 27.49 | 82.47 | high |
| | | Velocity (m/s) | 0.90±0.09 | 0.90±0.08 | 0.939 (0.882 to 0.966) | 1.91 (1.44 to 2.83) | 0.020 | 0.023 | 0.016 | 0.05 | moderate |
| | | Power (W) | 1188±218 | 1198±205 | 0.972 (0.930 to 0.989) | 2.28 (1.73 to 3.38) | 35.39 | 35.59 | 42.30 | 126.91 | high |
| 200 | Peak | Force (N) | 1560±168 | 1570±169 | 0.958 (0.897 to 0.984) | 1.77 (1.34 to 2.63) | 34.50 | 34.50 | 33.68 | 101.03 | moderate |
| | | Velocity (m/s) | 1.28±0.09 | 1.30±0.10 | 0.845 (0.647 to 0.937) | 2.40 (1.81 to 3.56) | 0.037 | 0.037 | 0.019 | 0.06 | moderate |
| | | Power (W) | 967±169 | 976±168 | 0.926 (0.820 to 0.971) | 3.92 (2.96 to 5.80) | 45.82 | 46.54 | 33.69 | 101.06 | moderate |
| | Mean | Force (N) | 1402±146 | 1405±146 | 0.957 (0.929 to 0.974) | 0.64 (0.48 to 0.95) | 30.26 | 11.93 | 29.19 | 87.57 | high |
| | | Velocity (m/s) | 0.82±0.08 | 0.83±0.08 | 0.929 (0.882 to 0.958) | 2.82 (2.13 to 4.17) | 0.020 | 0.027 | 0.016 | 0.05 | moderate |
| | | Power (W) | 1152±204 | 1171±208 | 0.943 (0.862 to 0.978) | 3.46 (2.61 to 5.11) | 49.13 | 48.38 | 41.16 | 123.49 | moderate |

BW, body weight; ICC, intraclass correlation coefficient; CV, coefficient of variation; SEM, standard error of measurement; TE, typical error; SWC, small worthwhile change; MWC, moderate worthwhile change.

demonstrated acceptable intra-session reliability in measuring all force-velocity-power variables. In contrast, some variables derived from vertical jump assessments exhibited unacceptable reliability. These results confirm our first hypothesis and partially support our second. Consequently, HDLPD can assess some force, velocity, and power variables during maximal leg extension with higher intrasession reliability than vertical jumps.

**Table 2. Intra-session reliability of force-velocity-power variables obtained from squat jump (SJ) and countermovement jump (CMJ).**

| Jump | Load condition | Type of value | Variables | Trial 1 | Trial 2 | ICC (95% CI) | CV (95% CI) | SEM | TE | SWC | MWC | Sensitivity |
|---|---|---|---|---|---|---|---|---|---|---|---|---|
| SJ | UL | Peak | Force (N) | 1440±141 | 1442±156 | 0.973 (0.931 to 0.989) | 1.19 (0.89 to 1.76) | 24.40 | 25.14 | 29.71 | 89.12 | high |
| | | | Velocity (m/s) | 2.66±0.15 | 2.62±0.16 | 0.750 (0.468 to 0.895) | 2.33 (1.76 to 3.44) | 0.078 | 0.077 | 0.031 | 0.09 | moderate |
| | | | Power (W) | 3240±365 | 3188±411 | 0.901 (0.766 to 0.961) | 3.16 (2.38 to 4.67) | 122.0 | 120.00 | 77.61 | 232.82 | moderate |
| | | Mean | Force (N) | 999±76 | 979±91 | 0.774 (0.509 to 0.906) | 3.05 (2.31 to 4.52) | 39.76 | 38.81 | 16.73 | 50.18 | moderate |
| | | | Velocity (m/s) | 1.03±0.14 | 0.98±0.16 | 0.595 (0.223 to 0.819) | 7.72 (5.83 to 11.4) | 0.095 | 0.093 | 0.030 | 0.09 | low |
| | | | Power (W) | 1112±160 | 1055±187 | 0.588 (0.213 to 0.816) | 8.38 (6.33 to 12.3) | 111.2 | 108.96 | 34.67 | 104.02 | low |
| | ML | Peak | Force (N) | 1590±154 | 1607±156 | 0.965 (0.905 to 0.987) | 1.40 (1.06 to 2.07) | 29.02 | 26.88 | 31.03 | 93.09 | high |
| | | | Velocity (m/s) | 2.23±0.10 | 2.21±0.09 | 0.814 (0.586 to 0.924) | 1.50 (1.13 to 2.22) | 0.041 | 0.040 | 0.019 | 0.06 | moderate |
| | | | Power (W) | 3089±327 | 3080±325 | 0.938 (0.847 to 0.976) | 1.98 (1.49 to 2.93) | 81.10 | 83.03 | 65.15 | 195.45 | moderate |
| | | Mean | Force (N) | 1167±123 | 1167±116 | 0.948 (0.869 to 0.979) | 1.82 (1.37 to 2.69) | 27.17 | 28.01 | 23.83 | 71.49 | moderate |
| | | | Velocity (m/s) | 0.89±0.12 | 0.87±0.10 | 0.797 (0.555 to 0.916) | 4.60 (3.47 to 6.80) | 0.050 | 0.051 | 0.022 | 0.07 | moderate |
| | | | Power (W) | 1080±175 | 1064±160 | 0.809 (0.573 to 0.922) | 5.11 (3.86 to 7.55) | 73.28 | 74.15 | 33.54 | 100.62 | moderate |
| | HL | Peak | Force (N) | 1758±197 | 1784±192 | 0.960 (0.884 to 0.985) | 1.85 (1.40 to 2.74) | 38.88 | 35.47 | 38.89 | 116.67 | high |
| | | | Velocity (m/s) | 1.81±0.10 | 1.86±0.09 | 0.692 (0.255 to 0.879) | 2.39 (1.80 to 3.53) | 0.053 | 0.047 | 0.019 | 0.06 | moderate |
| | | | Power (W) | 2793±342 | 2904±322 | 0.833 (0.508 to 0.939) | 4.09 (3.09 to 6.05) | 135.6 | 117.77 | 66.40 | 199.19 | moderate |
| | | Mean | Force (N) | 1317±145 | 1333±149 | 0.978 (0.925 to 0.992) | 1.28 (0.96 to 1.89) | 21.76 | 19.38 | 29.35 | 88.05 | high |
| | | | Velocity (m/s) | 0.72±0.09 | 0.75±0.08 | 0.696 (0.375 to 0.870) | 5.58 (4.22 to 8.26) | 0.046 | 0.046 | 0.017 | 0.05 | moderate |
| | | | Power (W) | 962±143 | 1007±146 | 0.788 (0.495 to 0.915) | 5.62 (4.24 to 8.31) | 66.55 | 61.48 | 28.91 | 86.73 | moderate |
| CMJ | UL | Peak | Force (N) | 1399±154 | 1396±138 | 0.950 (0.874 to 0.980) | 1.85 (1.40 to 2.74) | 32.62 | 33.57 | 29.18 | 87.53 | moderate |
| | | | Velocity (m/s) | 2.75±0.15 | 2.74±0.17 | 0.815 (0.580 to 0.925) | 1.38 (1.04 to 2.04) | 0.069 | 0.072 | 0.032 | 0.10 | moderate |
| | | | Power (W) | 3201±347 | 3209±345 | 0.864 (0.679 to 0.945) | 2.44 (1.84 to 3.61) | 127.4 | 130.54 | 69.13 | 207.39 | moderate |
| | | Mean | Force (N) | 1143±124 | 1140±109 | 0.938 (0.847 to 0.976) | 1.78 (1.35 to 2.64) | 29.03 | 29.74 | 23.32 | 69.97 | moderate |
| | | | Velocity (m/s) | 1.53±0.12 | 1.52±0.13 | 0.778 (0.512 to 0.908) | 2.71 (2.05 to 4.01) | 0.060 | 0.061 | 0.026 | 0.08 | moderate |
| | | | Power (W) | 1631±232 | 1615±200 | 0.831 (0.615 to 0.931) | 3.52 (2.66 to 5.20) | 88.76 | 90.41 | 43.19 | 129.56 | moderate |
| | ML | Peak | Force (N) | 1572±140 | 1566±170 | 0.947 (0.869 to 0.979) | 1.83 (1.38 to 2.71) | 35.77 | 36.68 | 31.08 | 93.23 | moderate |
| | | | Velocity (m/s) | 2.29±0.09 | 2.26±0.10 | 0.845 (0.507 to 0.946) | 1.35 (1.02 to 2.00) | 0.037 | 0.032 | 0.019 | 0.06 | moderate |
| | | | Power (W) | 3151±296 | 3091±331 | 0.922 (0.780 to 0.971) | 2.18 (1.65 to 3.22) | 87.43 | 79.17 | 62.61 | 187.84 | moderate |
| | | Mean | Force (N) | 1289±118 | 1278±129 | 0.971 (0.924 to 0.989) | 1.16 (0.87 to 1.71) | 20.99 | 20.06 | 24.66 | 73.97 | high |
| | | | Velocity (m/s) | 1.25±0.08 | 1.21±0.11 | 0.761 (0.385 to 0.908) | 2.81 (2.12 to 4.15) | 0.047 | 0.042 | 0.019 | 0.06 | moderate |
| | | | Power (W) | 1514±144 | 1466±192 | 0.852 (0.601 to 0.944) | 3.19 (2.41 to 4.72) | 64.63 | 58.88 | 33.60 | 100.81 | moderate |
| | HL | Peak | Force (N) | 1737±180 | 1756±180 | 0.954 (0.885 to 0.982) | 1.91 (1.44 to 2.82) | 38.62 | 37.15 | 36.02 | 108.05 | moderate |
| | | | Velocity (m/s) | 1.91±0.11 | 1.92±0.11 | 0.732 (0.426 to 0.888) | 2.06 (1.56 to 3.06) | 0.056 | 0.057 | 0.022 | 0.07 | moderate |
| | | | Power (W) | 2951±317 | 2985±319 | 0.839 (0.634 to 0.934) | 3.22 (2.44 to 4.77) | 127.5 | 128.40 | 63.57 | 190.72 | moderate |
| | | Mean | Force (N) | 1413±141 | 1419±141 | 0.971 (0.927 to 0.988) | 1.47 (1.11 to 2.18) | 24.00 | 24.37 | 28.20 | 84.59 | high |
| | | | Velocity (m/s) | 1.00±0.08 | 1.00±0.09 | 0.752 (0.459 to 0.897) | 3.47 (2.62 to 5.13) | 0.043 | 0.045 | 0.018 | 0.05 | moderate |
| | | | Power (W) | 1346±142 | 1352±147 | 0.793 (0.536 to 0.915) | 3.95 (2.98 to 5.85) | 65.81 | 67.20 | 28.93 | 86.79 | moderate |

UL, unloaded; ML, moderate-loaded; HL, high-loaded; ICC, intraclass correlation coefficient; CV, coefficient of variation; SEM, standard error of measurement; TE, typical error; SWC, small worthwhile change; MWC, moderate worthwhile change.

As shown in Table 1, the current study showed that the ddrobotec® System ELITE leg press device can assess these variables with high intra-session reliability (ICC = 0.845 to 0.974 and CV = 0.43% to 4.52%). On the other hand, some previous studies [4,11] examined the reliability of the Keiser leg press machine, an HDLPD. Infante et al. [11] reported that leg press one-repetition maximum in elderly women measured with a Keiser pneumatic resistance machine had excellent inter-session reliability (ICC = 0.972, CV = 6.32%). Furthermore, Larsen et al. [4] indicated that one repetition maximum estimated from the load-velocity relationship measured using a Keiser leg press machine showed almost perfect intersession

reliability (ICC = 0.99, CV = 3.0%). Although the resistance mechanism used by Keiser and ddrobotec® System ELITE (pneumatic resistance vs. pneumatic artificial muscles, respectively) is different, the current study suggested that such differences may not significantly affect the reliability of the maximal effort leg press performance variables. Therefore, HDLPD is considered a suitable tool for reliably assessing maximal leg extension performance, regardless of the type of load.

At the same load (i.e., 100% BW), the HDLPD showed similar or higher intrasession reliability for measuring force-velocity-power metrics than vertical jumps (Tables 1 and 2). These results are consistent with the study by Lindberg et al. [3], who reported more reliable force-velocity profiling with a Keiser leg press device than with SJ and CMJ. One possible factor contributing to HDLPD's high reliability is the reduction in participants' biological variability, likely due to the lower stability demands of the task. Because the participant's trunk was fixed to the seat during leg extension movements in the HDLPD, the trunk muscles were less likely to require significant activity. Indeed, Saeterbakken et al. [24] reported that the core muscles are less active in leg press exercises with a fixed trunk than in squat exercises. This reduction in task difficulty, represented by lower muscle activation demands, may have allowed for a more reproducible leg extension performance. Additionally, in the HDLPD, the range of motion during leg extension is fixed by the device for each trial, whereas in vertical jump movements, some degree of variation in leg extension range of motion is inevitable [3]. This restriction imposed by the HDLPD may have further contributed to minimizing biological variability among participants.

All force-velocity-power variables measured in the unloaded CMJ (i.e., body weight) had acceptable reliability (ICC > 0.75, CV < 10%). However, unacceptable relative reliability (ICC < 0.75) was observed for some variables (i.e., mean velocity and mean power) measured in the unloaded SJ (Table 2). In SJ, the participants are required to hold a half-squat position isometrically for a few seconds and not use countermovement before starting the jump. As participants often unintentionally incorporate small amplitude countermovement during SJ [25], the strict prohibition of such movements may have significantly increased the cognitive load of the task. Additionally, considering a previous study suggesting that increased cognitive load due to increased tasking decreases jumping performance [26], the difficulty of the SJ task may have contributed to between-trial performance variability and unreliability of the measured variables. Given the above, for a more reliable leg extension performance measurement, the CMJ may be more appropriate than the SJ, and the HDLPD may be more appropriate than the CMJ.

In the measurements using the HDLPD (Table 1), the mean force had a high sensitivity (i.e., TE < SWC) under all experimental loading conditions (i.e., 100% to 200% BW). Additionally, the mean power measured using the HDLPD under certain loading conditions (i.e., 160% and 180% BW) was also highly sensitive. However, other variables derived from the HDLPD were moderately sensitive. Therefore, for reliable performance measurement with HDLPD, the peak force or peak power may be the optimal outcome variable.

This study suggests that HDLPD is more useful than the vertical jump in reliably measuring force-velocity-power variables. The study, however, also has some limitations. The experimental sample for this study consisted of male field hockey players and was not large enough to extrapolate the data obtained to the entire population or other sports. Consequently, the results may not be applicable to female athletes or to sports that demand greater vertical force production, such as basketball. Given the lack of research on HDLPD, future studies should further test the reliability of the variables obtained from HDLPD in other populations. Additionally, the definitions of range of motion differed for HDLPD and vertical jumps, which may have affected the reliability results. However, given the movement characteristics and devices used, it was necessary to select appropriate definitions for the start and end of the HDLPD, and vertical jump movements. Furthermore, no direct comparison with other HDLPDs (e.g., the Keiser leg press machine) was conducted in this study. Therefore, it should be noted that the present results may not be completely generalizable to other HDLPDs. Further studies are required to verify the reliability of various HDLPDs.

## Conclusions

This study demonstrated that the HDLPD exhibits high intrasession reliability in assessing force-velocity-power variables during maximal leg extension efforts. Moreover, for certain variables, the HDLPD provided more reliable measurements

than traditional vertical jump assessments. These findings suggest that the HDLPD is a valuable tool for practitioners seeking a reliable method to evaluate lower-limb strength, speed, and power.

## Supporting information

**S1 Dataset. Original data.**
(XLSX)

## Acknowledgments

We would like to thank all the participants for their time and effort in this study and the staff of the Institute of Physical Education at Keio University for providing technical support.

## Author contributions

**Conceptualization:** Takuya Nishioka, Shota Yamaguchi, Takayuki Inami.

**Data curation:** Takuya Nishioka, Shota Yamaguchi, Daiki Hajima.

**Formal analysis:** Takuya Nishioka.

**Funding acquisition:** Takuya Nishioka, Shota Yamaguchi, Takayuki Inami.

**Investigation:** Takuya Nishioka, Shota Yamaguchi, Yasuhiro Kunita.

**Methodology:** Takuya Nishioka, Shota Yamaguchi, Takayuki Inami.

**Project administration:** Takuya Nishioka, Takayuki Inami.

**Resources:** Takuya Nishioka, Takayuki Inami.

**Software:** Takuya Nishioka.

**Supervision:** Takayuki Inami.

**Validation:** Takuya Nishioka.

**Visualization:** Takuya Nishioka.

**Writing – original draft:** Takuya Nishioka.

**Writing – review & editing:** Takuya Nishioka, Takayuki Inami.

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
