## [Decision Letter · Decision Letter 0]

9 Jul 2025

Dear Dr. Inami,

Thank you for submitting your manuscript to PLOS ONE. After careful consideration, we feel that it has merit but does not fully meet PLOS ONE’s publication criteria as it currently stands. Therefore, we invite you to submit a revised version of the manuscript that addresses the points raised during the review process.

We look forward to receiving your revised manuscript.

Kind regards,

Julio Alejandro Henriques Castro da Costa

Academic Editor

PLOS ONE

2. Please include a copy of Table 1 and 2 which you refer to in your text on page 11 and 12.

Additional Editor Comments (if provided):

Reviewers' comments:

Reviewer's Responses to Questions

**Comments to the Author**

1. Is the manuscript technically sound, and do the data support the conclusions?

Reviewer #1: Yes

Reviewer #2: Yes

2. Has the statistical analysis been performed appropriately and rigorously?

Reviewer #1: Yes

Reviewer #2: Yes

3. Have the authors made all data underlying the findings in their manuscript fully available?

Reviewer #1: Yes

Reviewer #2: Yes

4. Is the manuscript presented in an intelligible fashion and written in standard English?

Reviewer #1: Yes

Reviewer #2: Yes

Reviewer #1: The manuscript is well-structured, coherent, and addresses a relevant topic in the context of neuromuscular performance assessment. The proposed HDLPD method appears promising, and the results indicate good reliability. To further strengthen the manuscript and support the conclusions drawn, I recommend a few minor additions:

A brief mention justifying the sample size, such as through a power analysis, would be useful, even though the number of participants appears reasonable in the context of a reliability study.

It might also be helpful to note potential limitations regarding the generalizability of the results, as the sample consists exclusively of elite male athletes.

The comparison between the reliability of the HDLPD method and other tests (e.g., jump-based tests) is interesting but is not supported by a direct statistical comparison of the ICC coefficients. While not essential, adding such an analysis or phrasing the claims more cautiously could strengthen the argument.

Regarding the conclusion related to the safety of the HDLPD method, the current formulation is persuasive but relies more on theoretical reasoning than on direct measurements. A more reserved formulation or an additional reference could improve clarity.

These are minor adjustments and do not affect the overall validity of the study. However, they could enhance the manuscript’s coherence and scientific rigor. Congratulations to the authors on an interesting and well-conducted study.

Reviewer #2: 1. Methodological Enhancements Required

(1) Device calibration protocols must be explicitly stated in the Methods section. Specify the sampling frequency of force plates (e.g., 1000 Hz) and the cutoff frequency for low-pass filtering (e.g., 20 Hz Butterworth filter).

(2) Clarify the rationale for selecting 100-200% body weight loads. Provide reference to pilot testing or existing literature supporting this range for hockey athletes.

(3) Detail the force-velocity curve fitting algorithm. State whether linear or polynomial regression was applied and justify the choice.

2. Statistical Reporting Improvements

(1) Table 1 should include both absolute (SEM in Newtons) and relative (CV%) reliability metrics for force variables to comply with sports science reporting standards.

(2) Figure 2 needs error bars for individual data points when displaying mean force-velocity profiles across loads.

(3) Report exact p-values rather than thresholds (e.g., p=0.032 instead of p<0.05) for all pairwise comparisons.

3. Discussion Limitations to Address

(1) The discussion overstates HDLPD's superiority without biomechanical evidence. Modify claims to reflect that reliability differences may stem from joint angle specificity (knee flexion at 90° in HDLPD vs. variable angles in jumps) rather than inherent device superiority.

(2) Acknowledge the sample limitation by adding: "Results may not generalize to female athletes or sports requiring greater vertical force production (e.g., basketball)."

4. Ethical Compliance Note

(1) Ensure the Ethics Statement includes the institutional review board approval number and date (currently missing in Methods).

(2) Confirm in the Data Availability Statement whether raw force-time data will be shared via repository (e.g., Figshare) or available upon request.

**Do you want your identity to be public for this peer review?** For information about this choice, including consent withdrawal, please see our Privacy Policy

Reviewer #1: No

Reviewer #2: No

---

## [Author Response · Author response to Decision Letter 1]

31 Jul 2025

Thank you for the opportunity to resubmit our manuscript entitled, “Comparison of intra-session reliability of force-velocity-power variables between a horizontal dynamic leg press device and vertical jump tests.” We have carefully reviewed the reviewers’ comments and revised the manuscript accordingly. All necessary changes have been made, and detailed responses to each comment are provided below. We hope the revised manuscript now meets the criteria for publication in your journal.

Reviewer #1: The manuscript is well-structured, coherent, and addresses a relevant topic in the context of neuromuscular performance assessment. The proposed HDLPD method appears promising, and the results indicate good reliability. To further strengthen the manuscript and support the conclusions drawn, I recommend a few minor additions:

Answer: Thank you for your valuable comments. Please find our detailed responses to each of your suggestions below.

A brief mention justifying the sample size, such as through a power analysis, would be useful, even though the number of participants appears reasonable in the context of a reliability study.

Answer: As you suggested, we have added an explanation regarding the sample size justification and included an appropriate reference, as follows:

“Sample size calculations were performed using G*Power software (version 3.1.9.6, Düsseldorf, Germany) to determine the number of participants required to assess reliability [12]. Based on a detected reliability of 0.9, a minimal acceptable reliability of 0.5, a significance level of 0.05, and a power of 0.95, the required sample size was calculated to be 17 participants. Accordingly, 19 male field hockey players (age: 19.6 ± 1.1 years, height: 173.0 ± 5.9 cm, body mass [BM]: 61.8 ± 6.5 kg, mean ± SD) were recruited to participate in this study” (lines 77–82)

It might also be helpful to note potential limitations regarding the generalizability of the results, as the sample consists exclusively of elite male athletes.

Answer: As you suggested, we have added a statement regarding the limitation of the study's generalizability. (lines 259–261)

The comparison between the reliability of the HDLPD method and other tests (e.g., jump-based tests) is interesting but is not supported by a direct statistical comparison of the ICC coefficients. While not essential, adding such an analysis or phrasing the claims more cautiously could strengthen the argument.

Answer: To the best of our knowledge, statistical tests for directly comparing reliability metrics (e.g., ICCs) are not implemented in commonly used statistical software such as SPSS, nor have they been widely applied in previous studies. Therefore, we compared the reliability between the HDLPD and vertical jump tests based on the interpretation of ICC values and sensitivity. Specifically, we considered an ICC greater than 0.90 as indicating excellent reliability, which we interpreted as higher than a “good” ICC ranging from 0.75 to 0.90. This approach to interpreting and comparing ICC values is consistent with those used in previous studies (1, 2).

1. Katoh M, Yamasaki H. Comparison of reliability of isometric leg muscle strength measurements made using a hand‑held dynamometer with and without a restraining belt. J Phys Ther Sci. 2009;21(1):37‑42. doi:10.1589/jpts.21.37.

2. Wei X, Zheng X, Zhu H, et al. Comparison of Intrasession Reliability and Sensitivity Across Different Deceleration-Test Results in Male and Female Soccer Players. Int J Sports Physiol Perform. 2025;20(6):779-785. doi:10.1123/ijspp.2024-0432.

Regarding the conclusion related to the safety of the HDLPD method, the current formulation is persuasive but relies more on theoretical reasoning than on direct measurements. A more reserved formulation or an additional reference could improve clarity.

Answer: As you suggested, we have removed the description regarding the safety of the HDLPD from the manuscript. (lines 33 and 276)

These are minor adjustments and do not affect the overall validity of the study. However, they could enhance the manuscript’s coherence and scientific rigor. Congratulations to the authors on an interesting and well-conducted study.

Answer: Thank you for your valuable comment. Your feedback was invaluable in strengthening our manuscript. We hope that the revisions meet your expectations.

Reviewer #2: 1. Methodological Enhancements Required

(1) Device calibration protocols must be explicitly stated in the Methods section. Specify the sampling frequency of force plates (e.g., 1000 Hz) and the cutoff frequency for low-pass filtering (e.g., 20 Hz Butterworth filter).

Answer: The sampling frequency of force plates and the HDLPD were 1,000 and 200 Hz, respectively, and the cutoff frequency for low-pass filtering was 50 Hz, as described in the Materials and methods section. For the HDLPD, the upper limit of the sampling rate was relatively low (200 Hz), and the raw data exhibited minimal noise. Given this characteristic, a low-pass filter was not applied to the HDLPD data to avoid over-smoothing. Accordingly, the cutoff frequency for the HDLDP was not specified.

(2) Clarify the rationale for selecting 100-200% body weight loads. Provide reference to pilot testing or existing literature supporting this range for hockey athletes.

Answer: In our pilot testing, all participants were able to complete the leg press movement with a 200% body weight (BW) load. Furthermore, this load range corresponds to the typical load range observed in vertical jumps. Therefore, we selected 100–200% BW loads for the HDLPD in this study. We have added the following explanation to the manuscript:

“In our pilot testing, all participants successfully performed the leg press at 200% BW. Furthermore, this load range corresponds to the forces typically observed during vertical jumps [3]; therefore, 100–200% BW loads were selected for the HDLPD in the present study.” (lines 110–113)

(3) Detail the force-velocity curve fitting algorithm. State whether linear or polynomial regression was applied and justify the choice.

Answer: We did not perform force–velocity curve fitting in the present study; instead, we reported the measured peak and mean values of force and velocity. Please refer to Tables 1 and 2.

2. Statistical Reporting Improvements

(1) Table 1 should include both absolute (SEM in Newtons) and relative (CV%) reliability metrics for force variables to comply with sports science reporting standards.

Answer: The SEM and CV values are already included; please refer to Tables 1 and 2.

(2) Figure 2 needs error bars for individual data points when displaying mean force-velocity profiles across loads.

Answer: As mentioned above, we did not create Figure 2 or include force-velocity profiles in the current study.

(3) Report exact p-values rather than thresholds (e.g., p=0.032 instead of p<0.05) for all pairwise comparisons.

Answer: Since reliability metrics (e.g., ICC and CV) cannot be directly compared using statistical tests, p-value–based statistical analysis was not performed.

3. Discussion Limitations to Address

(1) The discussion overstates HDLPD’s superiority without biomechanical evidence. Modify claims to reflect that reliability differences may stem from joint angle specificity (knee flexion at 90° in HDLPD vs. variable angles in jumps) rather than inherent device superiority.

Answer: As you suggested, we have added a discussion noting that differences in reliability may be attributable to variations in leg extension range of motion, as follows:

“Additionally, in the HDLPD, the range of motion during leg extension is fixed by the device for each trial, whereas in vertical jump movements, some degree of variation in leg extension range of motion is inevitable [3]. This restriction imposed by the HDLPD may have further contributed to minimizing biological variability among participants.” (lines 232–236)

(2) Acknowledge the sample limitation by adding: “Results may not generalize to female athletes or sports requiring greater vertical force production (e.g., basketball).”

Answer: In response to your suggestion, we have added the following sentence:

“Consequently, the results may not be applicable to female athletes or to sports that demand greater vertical force production, such as basketball.” (lines 259–261)

4. Ethical Compliance Note

(1) Ensure the Ethics Statement includes the institutional review board approval number and date (currently missing in Methods).

Answer: As you suggested, we have included the institutional review board approval number and date in the manuscript. (lines 89–90)

(2) Confirm in the Data Availability Statement whether raw force-time data will be shared via repository (e.g., Figshare) or available upon request.

Answer: We intend to provide the raw force–time data to readers upon request. This has been added to the data availability information.

---

## [Decision Letter · Decision Letter 1]

20 Aug 2025

Comparison of intra-session reliability of force-velocity-power variables between a horizontal dynamic leg press device and vertical jump tests

PONE-D-25-21984R1

Dear Dr. Inami,

We’re pleased to inform you that your manuscript has been judged scientifically suitable for publication and will be formally accepted for publication once it meets all outstanding technical requirements.

Kind regards,

Julio Alejandro Henriques Castro da Costa

Academic Editor

PLOS ONE

Additional Editor Comments (optional):

Reviewers' comments:

Reviewer's Responses to Questions

**Comments to the Author**

Reviewer #1: All comments have been addressed

Reviewer #2: All comments have been addressed

2. Is the manuscript technically sound, and do the data support the conclusions?

Reviewer #1: (No Response)

Reviewer #2: Yes

3. Has the statistical analysis been performed appropriately and rigorously?

Reviewer #1: (No Response)

Reviewer #2: Yes

4. Have the authors made all data underlying the findings in their manuscript fully available?

Reviewer #1: (No Response)

Reviewer #2: Yes

5. Is the manuscript presented in an intelligible fashion and written in standard English?

Reviewer #1: (No Response)

Reviewer #2: Yes

Reviewer #1: (No Response)

Reviewer #2: The manuscript presents a well-conducted reliability study with appropriate methodology and comprehensive statistical analysis. The authors have adequately addressed all reviewer concerns and the findings provide valuable practical insights for sports science practitioners. I recommend acceptance.

**Do you want your identity to be public for this peer review?** For information about this choice, including consent withdrawal, please see our Privacy Policy

Reviewer #1: No

Reviewer #2: No

---

## [Editor Report · Acceptance letter]

PONE-D-25-21984R1

PLOS ONE

Dear Dr. Inami,

I'm pleased to inform you that your manuscript has been deemed suitable for publication in PLOS ONE. Congratulations! Your manuscript is now being handed over to our production team.

Kind regards,

on behalf of

Dr. Julio Alejandro Henriques Castro da Costa

Academic Editor

PLOS ONE